# The use of laser lead extraction sheath in the presence of supra-cardiac occlusion of the central veins for cardiac implantable electronic device lead upgrade or revision

**Sameer Al-Maisary** ⬦*, **Gabriele Romano**◉, **Matthias Karck**◉, **Raffaele De Simone**◉, **Jamila Kremer**◉

Department of Cardiac Surgery, Heidelberg University Hospital, Heidelberg, Baden-Württemberg, Germany

◉ These authors contributed equally to this work.
* yemendoctor@yahoo.com

## Abstract

**Data Availability Statement:** All relevant data are within the paper and its Supporting Information files.

### Background

The implantation of cardiac implantable electronic devices (CIED) has increased in the last decades with improvement in the quality of life of patients with cardiac rhythm disorders. The presence of bilateral subclavian, innominate or superior vena cava obstruction is a major limitation to device revision and/or upgrade.

### Methods and material

This is retrospective study of patients who underwent laser-assisted lead extraction (LLE) (GlideLight laser sheath, Spectranetics Corporation, Colorado Springs, USA) with lead revision or upgrade using the laser sheath as a guide rail. Patients with known occlusion, severe stenosis or functional obstruction of the venous access vessels with indwelling leads were included in this study.

### Results

106 patients underwent percutaneous LLE with lead revision and/or upgrade. Preoperative known complete occlusion or severe stenosis of access veins was present in 23 patients (21.5%). More patients with implantable cardioverter-defibrillator (ICD) underwent LLE (64.1%) than patients with CRT-Ds (24.5%) and pacemaker patients (11.3%). In total 172 leads were extracted: 79 (45.9%) single-coil defibrillator leads, 35 (20.3%) dual-coil defibrillator leads, 31 (18.0%) right atrial leads, 24 (13.9%) right ventricular leads and three (1.7%) malfunctional coronary sinus left ventricular pacing leads. The mean age of leads was 99.2 ±65.6 months. The implantation of new leads after crossing the venous stenosis/obstruction was successful in 98 (92.4%) cases. Postoperative complications were pocket hematoma in two cases and wound infection in one case. No peri-operative and no immediate postoperative death was recorded. One intraoperative superior vena cava tear was treated by immediate thoracotomy and surgical repair.

**Funding:** The author(s) received no specific funding for this work.

**Competing interests:** The authors have declared that no competing interests exist.

## Conclusion

In a single-center study on LLE in the presence of supra-cardiac occlusion of the central veins for CIED lead upgrade and revision we could demonstrate a low procedural complication rate with no procedural deaths. Most of the leads could be completely extracted to revise or upgrade the system. Our study showed a low complication rate, with acceptable mortality rates.

## Introduction

The implantation of cardiac implantable electronic devices (CIED) has increased in the last decades due to beneficial survival rates with improvement in the quality of life of patients with cardiac rhythm disorders. Nevertheless, the widespread use is also associated with an increase in post-implantation complications. Infection and dysfunction are the most prominent ones due to their immediate involvement in therapy strategies and the life expectancy of the affected patients with a significant increase in morbidity and mortality [1]. A recent survey showed that 28% of cardiac resynchronization therapy device (CRT-D) implantations were performed in patients with pre-existing devices [2].

After the introduction of laser lead extraction (LLE), the prevalence has increased continuously. These are expensive and time-consuming procedures needing highly trained and experienced operators. The percutaneous lead extraction is correspondingly accompanied with rare but serious life-threatening complications [3,4]. Considering the recent developments in the domain of CIED, the necessity for system revision and upgrade is expected to rise in the future with an increasing number of LLEs to come.

The presence of asymptomatic ipsilateral or bilateral subclavian, innominate or superior vena cava obstruction is a major limitation to device revision and/or upgrade [4,18]. Many reports suggest that such an obstruction occurs in 30 to 50% of cases [4–8] and only 1–3% of the patients are symptomatic.

Herein, we describe our first results of 80 Hz high frequency laser extraction with lead revision or CIED upgrade using the laser sheath as a guide rail in patients with known occlusion or severe stenosis of the venous access vessels with indwelling nonfunctional leads. Patients with functional vein obstruction were also included.

## Methods and materials

This is a retrospective study of consecutive patients who underwent laser-assisted lead extraction (using GlideLight laser sheath, Spectranetics Corporation, Colorado Springs, USA) with lead revision or CIED upgrade using the laser sheath as a guide rail as seen in Figs 1–8. Patients who underwent the procedure between May 2010 and January 2020 in the department of cardiac surgery at Heidelberg University Hospital, Germany were included in this study. Patients' data were extracted from our database and included patients' demographics, comorbidities, device and lead type, reason for extraction, procedural success, details of procedure, intra- and postoperative complications and in-hospital mortality up to one year. In this cohort, patients were referred from external hospitals or from our electrophysiological outpatient clinic.

Patients with known occlusion or severe stenosis of the venous access vessels (identified using bilateral Venography) with indwelling nonfunctional leads were included as well as patients with functional vein obstruction as in Fig 1.

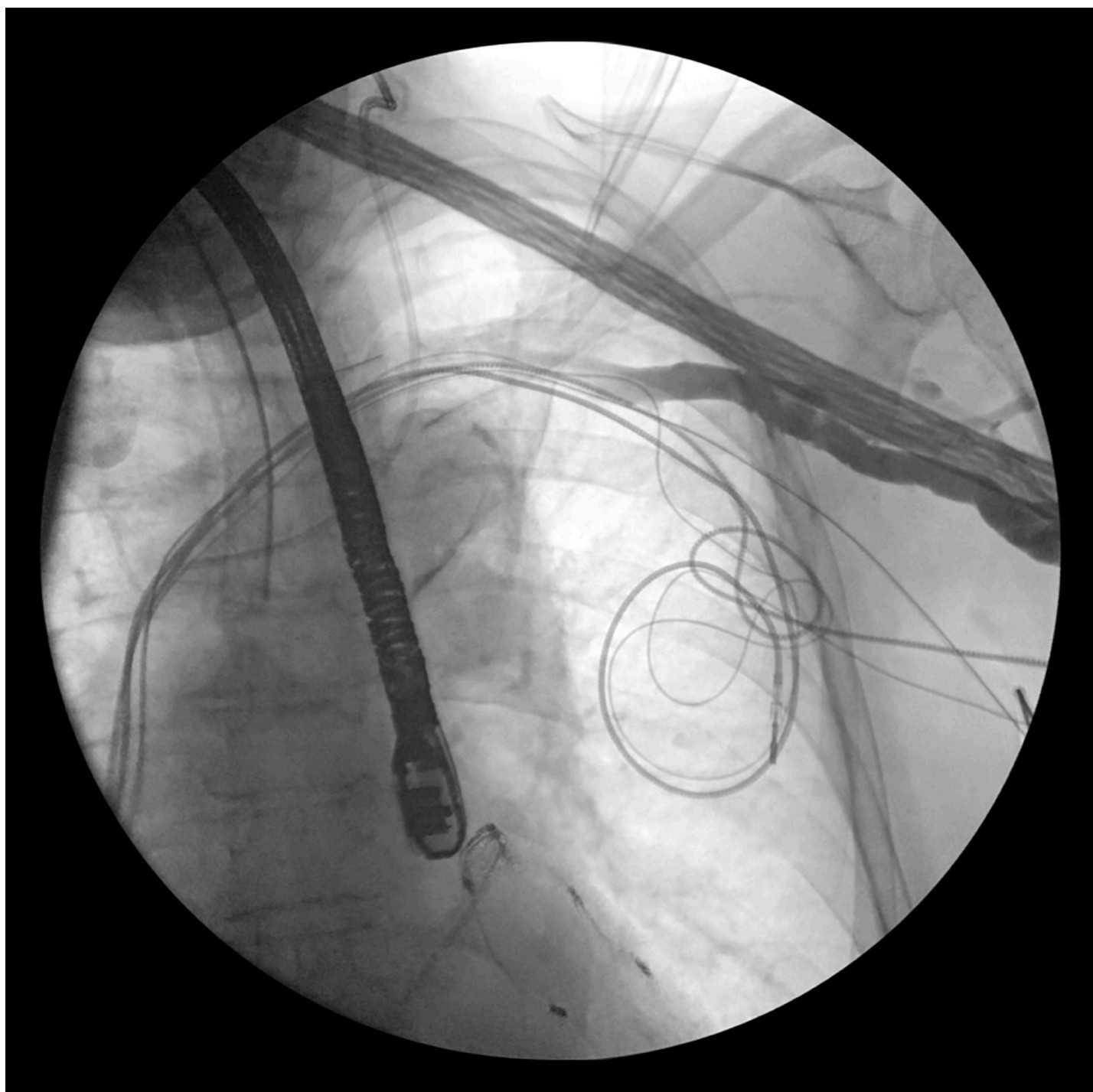

**Fig 1. Angiographic venous stenosis with indwelling nonfunctional leads.**

A functional obstruction was declared if intraoperatively no guide wire could cross the puncture site and when ipsilateral lead revision or upgrade was deemed impossible without laser lead extraction.

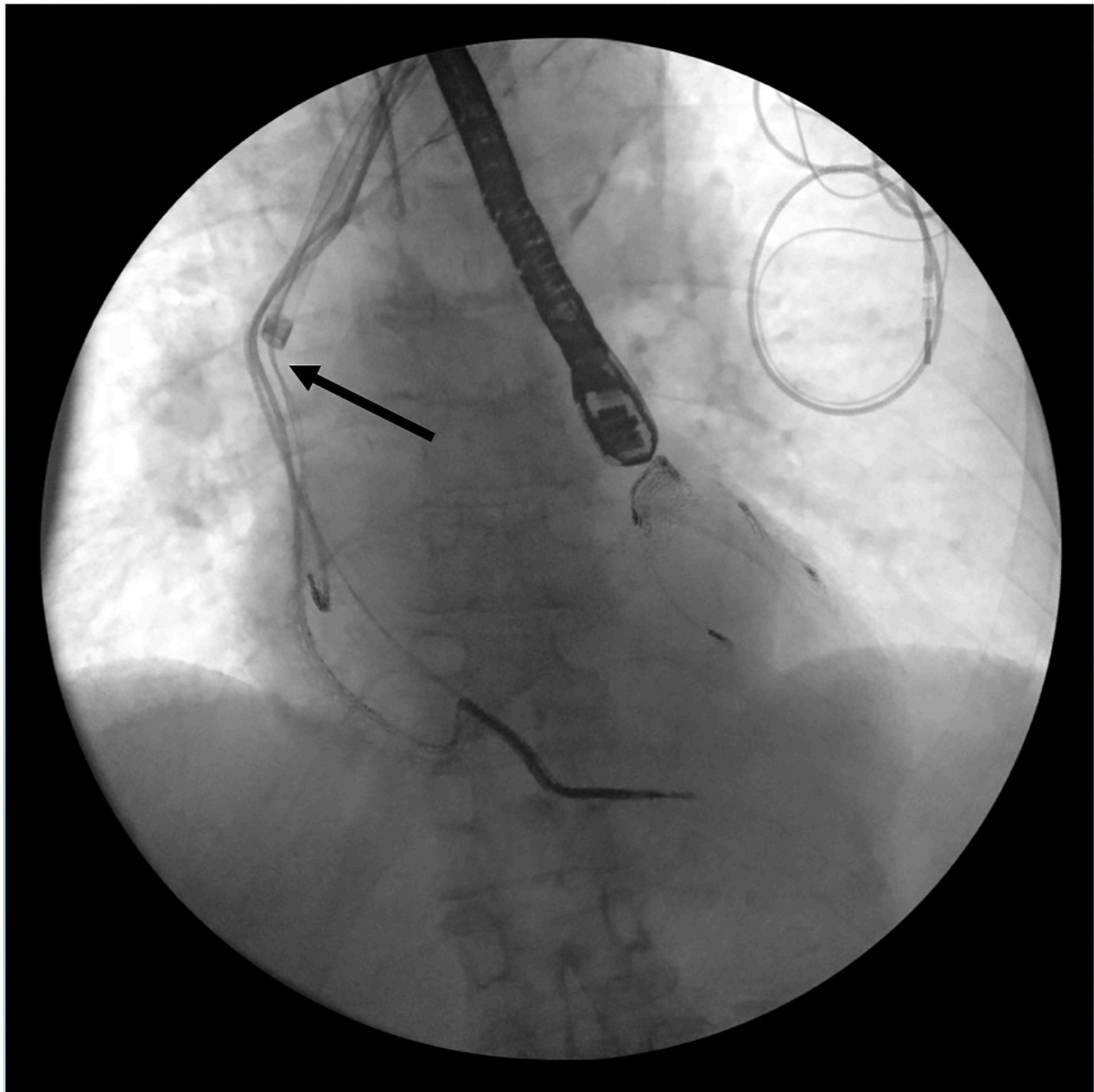

**Fig 2. Advancing of the laser sheath over the lead marked by the arrow.**

## Surgical procedure

All procedures were performed under general anesthesia with continuous arterial blood pressure monitoring. After opening the generator pocket, the ipsilateral subclavian vein was punctured, a guide wire was advanced towards the right atrium. If the guide wide could cross the

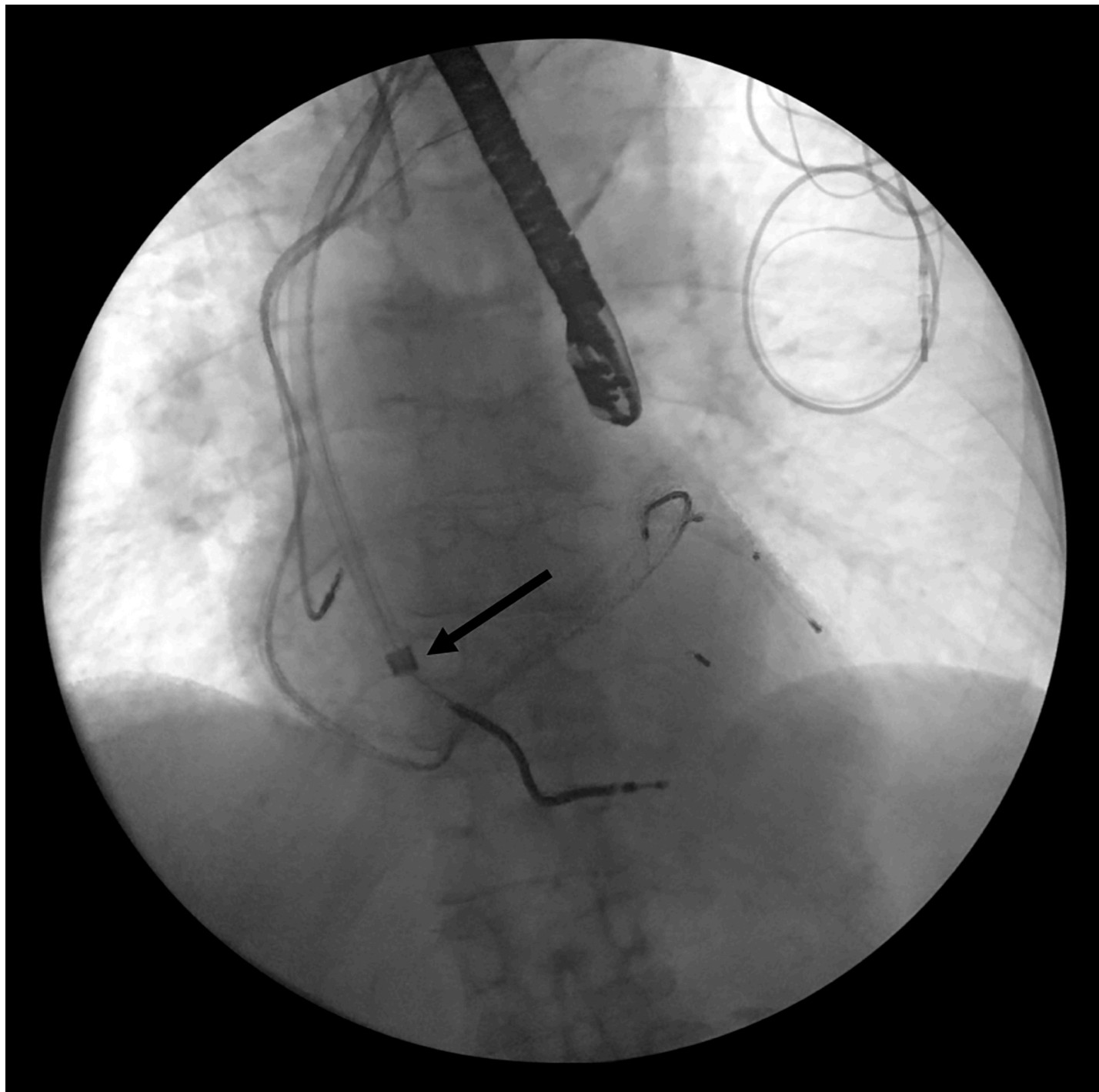

**Fig 3. Further advancing of the laser sheath over the lead marked by the arrow.**

subclavian/innominate vein, LLE initiated. Under fluoroscopic guidance, the lead extraction started by inserting a lead locking stylet into the inner coil lumen (LLD EZ lead locking device; Spectranetics) and was advanced until the lead tip then locked. A suture was tied around the insulation and the locking stylet. After that, the laser sheath (Glide Light 80 Hz, 14 or 16

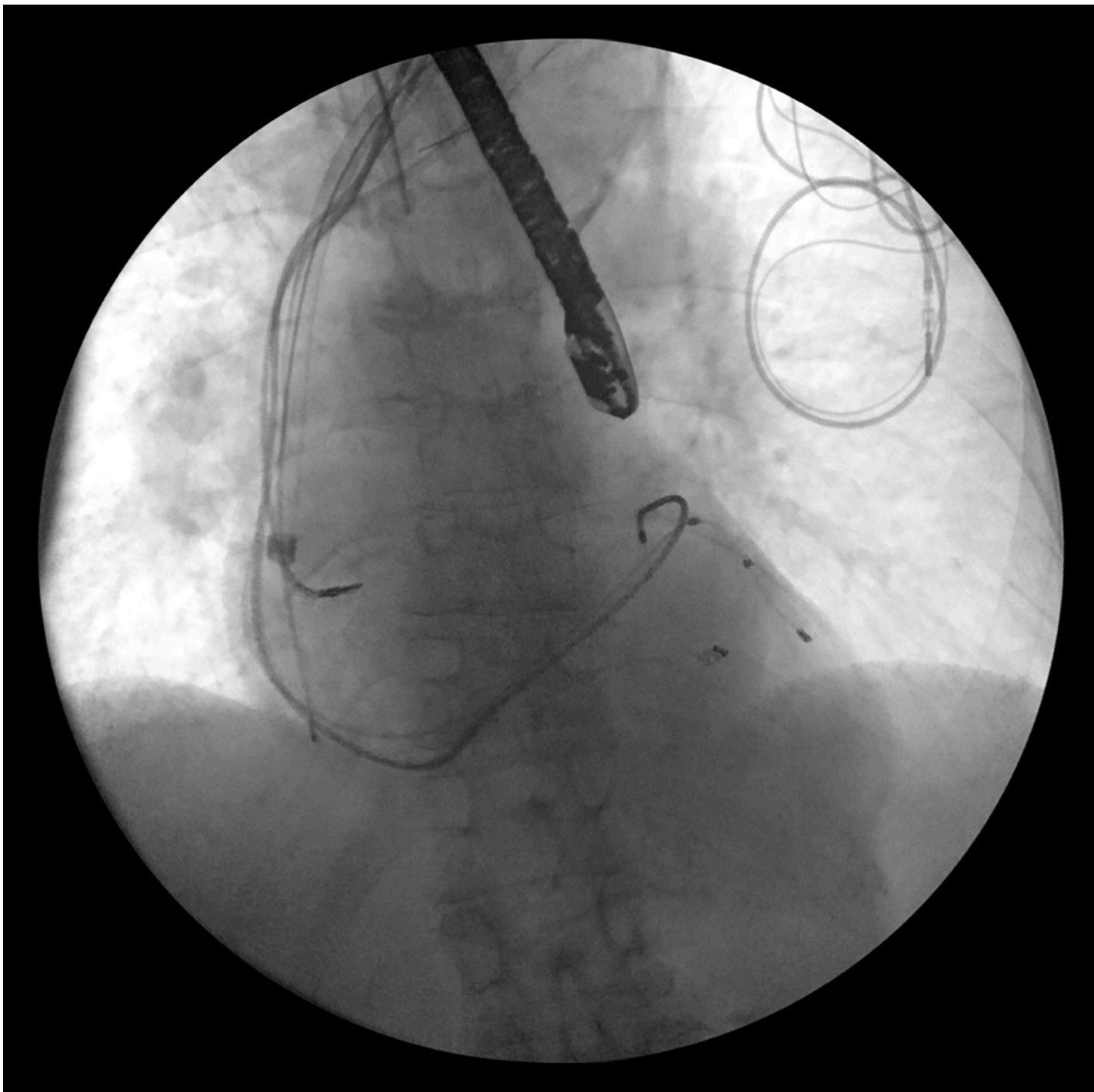

**Fig 4. Extraction of the lead through the laser sheath.**

French) was advanced over the lead until the locking stylet emerged from the other side of the laser sheath as in Figs 2 and 3.

Ultraviolet laser was then applied while gradually advancing the sheath over the lead under traction until the lead was freed (Fig 4).

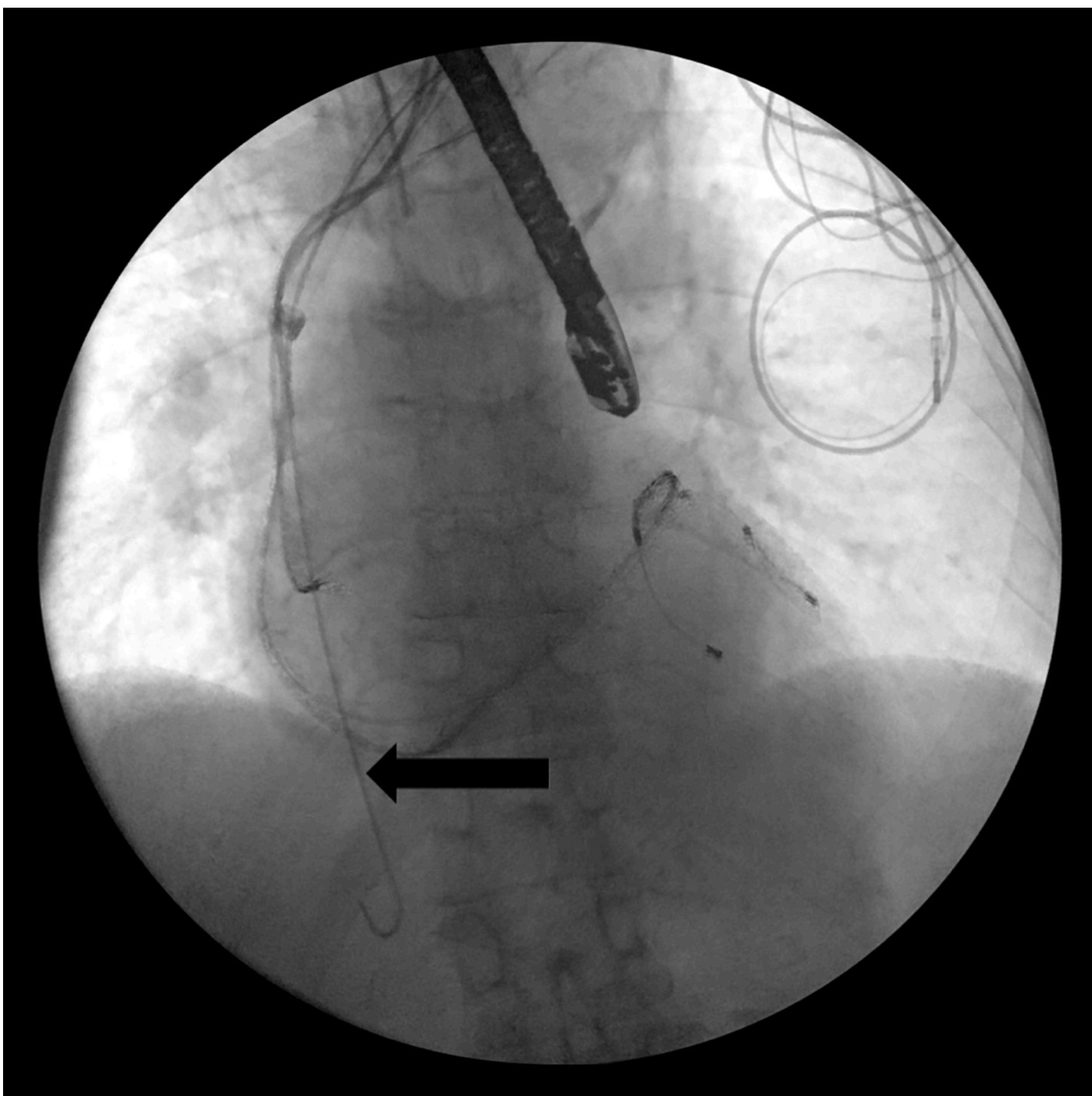

**Fig 5. Insertion of the guide wire (marked by the arrow) through the laser sheath for the implantation of a new lead over the venous stenosis.**

After that a guide wire (at least 100cm long) was passed through the laser sheath and the sheath was removed leaving the guide wire in place to maintain vascular access (Figs 5 and 6).

Any remaining ipsilateral non-functional leads were also extracted. Afterwards an introducer sheath was advanced over the guidewire into the venous system for new lead implantation (Figs 7 and 8).

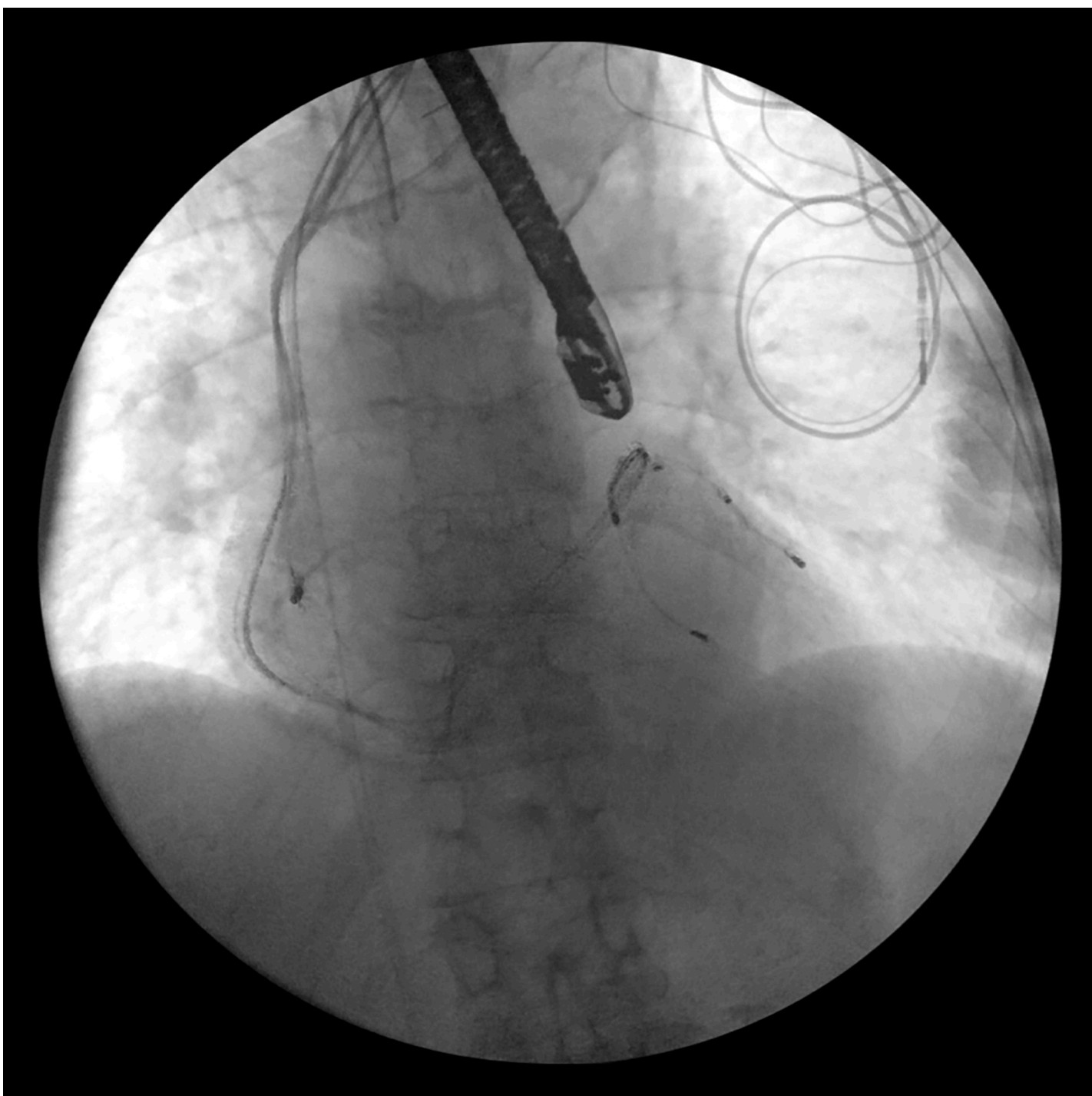

**Fig 6. Retraction of the laser sheath.**

Other guide wires were added depending on the number of the leads to be implanted. Procedural success and failure were defined according to the definitions of the 2017 Heart Rhythm Society and the 2018 European Heart Rhythm Association expert consensus. Removal

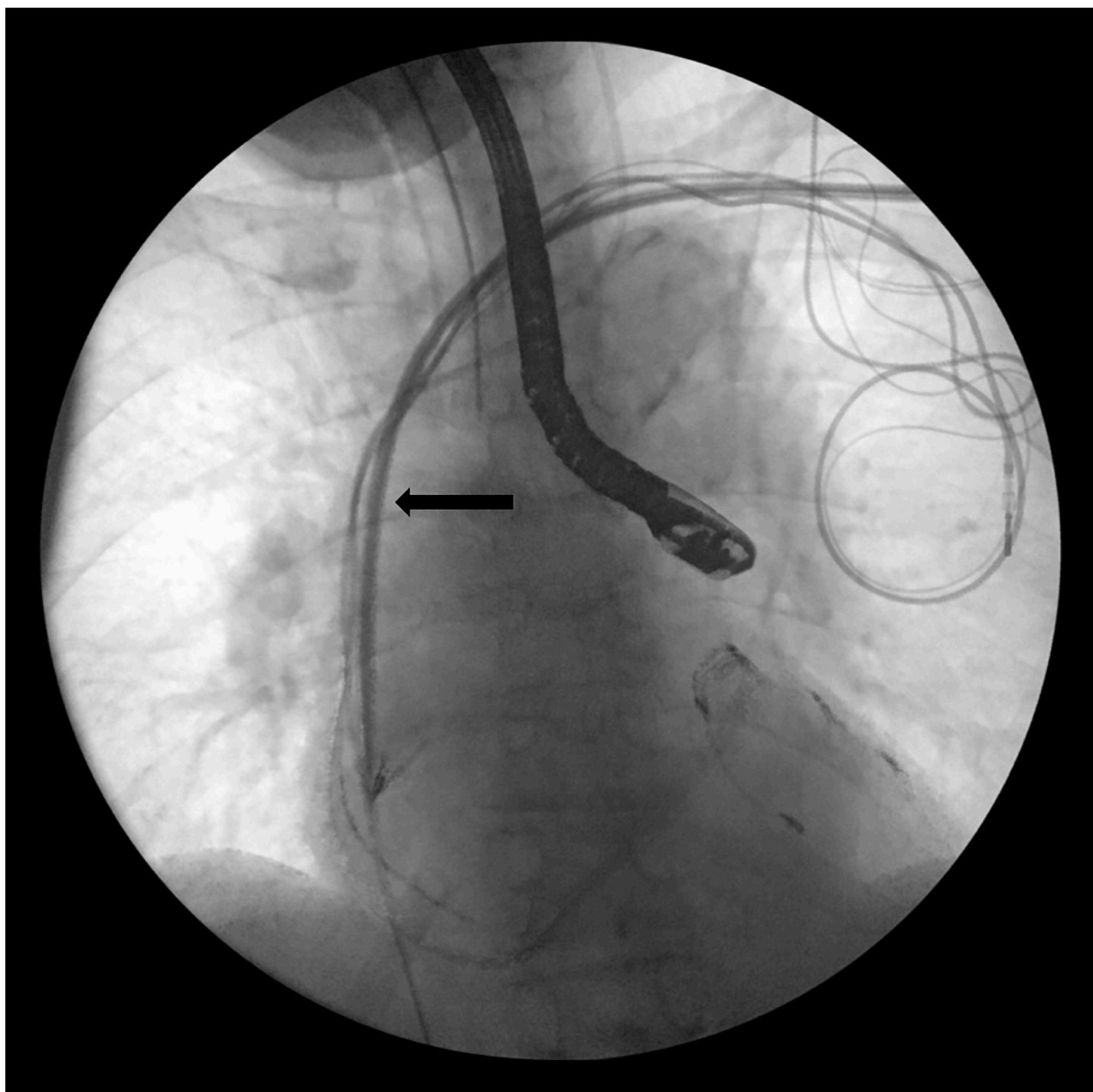

**Fig 7. Positioning of the introducer sheath (marked by the arrow) for lead implantation over the guide wire.**

of all non-functional ipsilateral leads with reimplantation of needed new leads was considered as procedural success. Removal of some leads was considered as partial success. Procedural failure was defined as the inability to re-implant any lead [5].

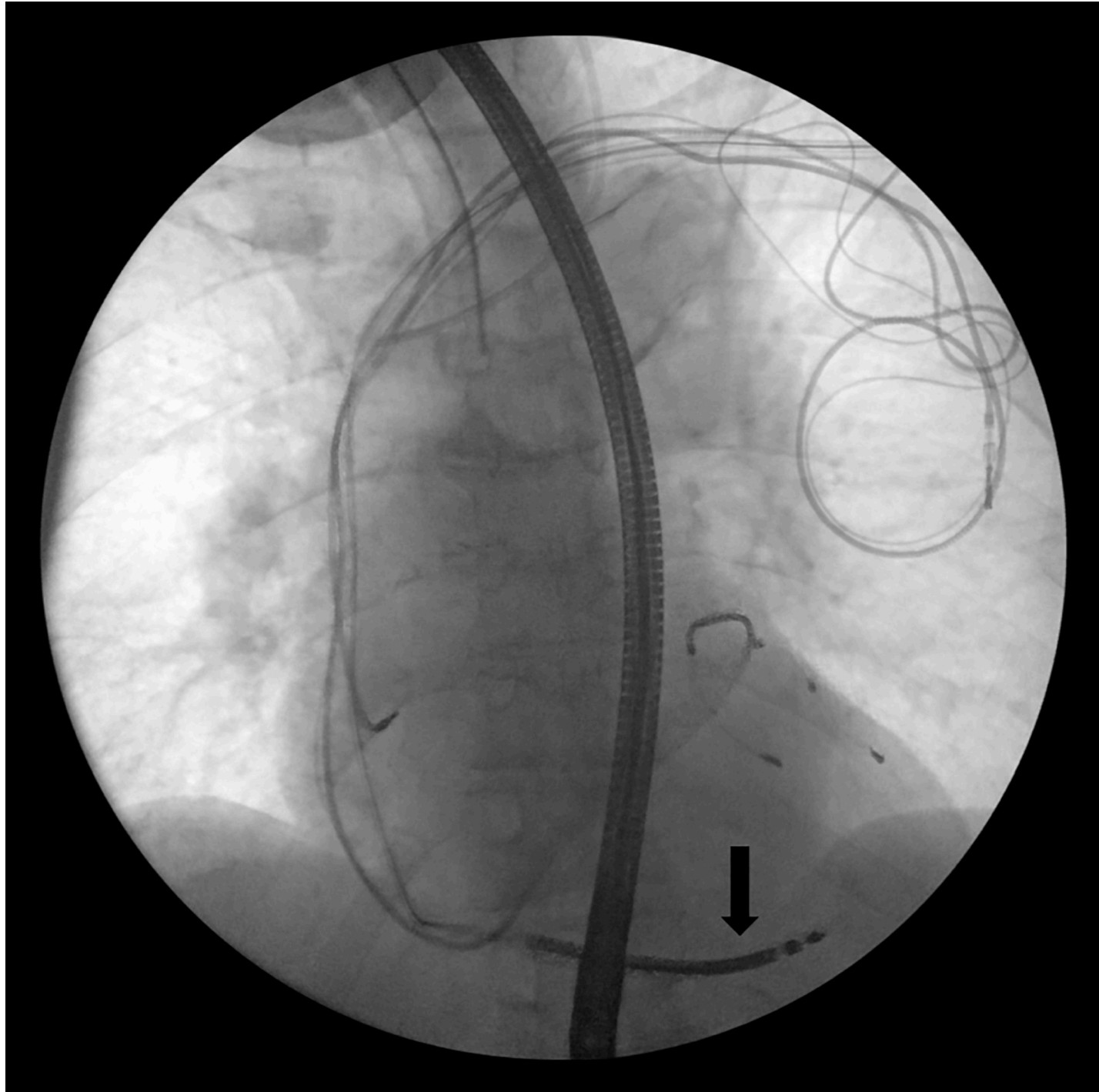

**Fig 8. Successful new lead implantation (marked by the arrow) in the right ventricle.**

This study complied with the Declaration of Helsinki and was approved by the ethical committee of the Medical University Heidelberg, S-597/2019. The need for consent was waived by the ethics committee. The data was for the purpose of data completion not anonym for the investigators.

**Table 1. Baseline characteristics.**

| Characteristic | |
|---|---|
| Age (years) | 59.2±15.9 |
| Male Sex, n. (%) | 63(58.9) |
| BMI | 28.2±14.4 |
| LVEF (%) | 39.4±14.2 |
| NYHA ≥III, n. (%) | 25(23.3) |
| IHD, n. (%) | 48(44.9) |
| HTN, n. (%) | 81(75.7) |
| Diabetes mellitus, n. (%) | 29(27.1) |
| Renal insufficiency, n. (%) | 9(8.4) |

BMI = body mass index; HTN = hypertension; IHD = ischemic heart disease; LVEF = left ventricular ejection fraction.

## Statistics

All statistical analyses were performed using the IBM SPSS Statistics version 25 software (SPSS, Chicago, IL). Normally distributed continuous variables were reported as mean ± standard deviation. Categorical variables were reported as frequencies and percentages.

## Results

From May 1, 2010 to January 1, 2020 a total of 106 patients underwent percutaneous laser lead extraction with lead revision and/or upgrade in our center. Preoperative known complete

**Table 2. Device characteristics.**

| Device | n (%) |
|---|---|
| Pacemaker | 12 (11) |
| ICD | 68 (64) |
| CRT-D | 26 (24) |
| Indication for extraction: | |
| Lead revision | 99 (93) |
| Lead upgrade | 3 (3) |
| Lead revision and upgrade | 4 (49 |
| Nature of upgrade | |
| Pacemaker to CRT-D | 2 (2) |
| Pacemaker to ICD | 3 (3) |
| ICD to CRT-D | 2 (2) |
| Number of extracted leads | |
| Total | 172 |
| Single coil RV-Lead | 79 (46) |
| Double coil RV-Lead | 35 (20) |
| RV-Lead | 24 (14) |
| RA-Lead | 31 (18) |
| CS-Lead | 3 (2) |
| Age of extracted leads (Months) | 99.22±65.6 |

ICD, implantable cardioverter defibrillator; CRT-D, cardiac resynchronization therapy-defibrillator; RV, right ventricle; RA, right atrium; CS, coronary sinus.

**Table 3. Procedure outcome.**

| Outcome | n (%) |
|---|---|
| Successful new lead implantation: | 98 (92.4) |
| Intraoperative complications | 1 (0,9) |
| Postoperative minor complications | 3 (2.8) 2x Pocket hematoma, 1x wound infection |
| Postoperative major complications | None |
| Intraoperative mortality | None |
| One-month mortality | None |
| One-year mortality | 4 (3.7) (2x cardiac decompensation,2x sepsis) |

occlusion or severe stenosis of access veins was present in 23 patients (21.5%) and was confirmed using Venography. The remainder showed intraoperative functional obstruction that prevented the crossing of guide wires into the right atrium. Main patient characteristics are seen in Table 1. The mean age of patients was years 59.2±15.9 and 63 patients were male (58.9%). The mean body mass index of the patients was 28.2±14.4 and the mean left ventricular ejection fraction was 39.4±14.2%. Twenty-five patients (23.3%) were in functional status NYHA ≥ III.

Patients with implantable cardioverter-defibrillator (ICD) underwent most of the extraction (64.1%) followed by patients with CRT-Ds (24.5%) and pacemaker patients (11.3%). The total number of extracted leads was 172. Of these, 79 (45.9%) single-coil defibrillator leads, 35 (20.3%) dual-coil defibrillator leads, 31 (18.0%) were right atrial leads, 24 (13.9%) right ventricular leads and three (1.7%) malfunctional coronary sinus left ventricular pacing leads. The mean age of leads was 99.2±65.6 months. Most of the patients underwent lead revision (99 patients, 74.5%) and three patients needed a simple system upgrade with another four patients needing system upgrade with revision of malfunctioning leads. Most of the leads were extracted due to malfunction (170) and only two functional right ventricular leads were extracted to upgrade the devices. Furthermore, 8 functional abandoned leads were included in this study as well as 99 malfunctioning abandoned leads. In 9 patients, lead extraction was only partially successful, with 16 leads remaining in situ after partial extraction. Device characteristics are mentioned in Table 2.

The implantation of new leads after crossing the venous stenosis/obstruction was successful in 98 (92.4%) cases. Failure was reported in eight cases. In seven cases the laser sheath could not be advanced beyond a second stenosis after freeing the lead tip. In one case, the lead fragmented prematurely preventing crossing the stenosis and therefore resulting in only partial extraction. All 8 patients with unsuccessful obstruction crossing received leads from the other non-stenosed subclavian vein and one patient received an additional epicardial left ventricular lead.

One patient suffered from superior vena cava tear intraoperatively and thoracotomy was performed. Postoperative complications were limited to pocket hematoma in two cases and wound infection in one case. No peri-operative and no immediate postoperative death was recorded. However, 30-day-mortaliy was 1.9% with two patients dying after 7 and 17 days. One suffered from acute bleeding of an intercostal artery with subsequent hemothorax with subsequent cardiac decompensation. The second patient died from sepsis after pneumonia and necrotic pancreatitis. Survival rate during follow-up was 86.8% with a cumulative follow-up time of 74693 days. The outcome of procedures is noted in Table 3.

## Discussion

We presented our results using the laser sheath as a rail for lead implantation after extraction of indwelling leads in the presence of ipsilateral venous obstruction to revise and /or upgrade Leads. This is the largest series, in which lead laser extraction is used to facilitate the lead revision and/or upgrade after the creation of a new tunnel using the laser sheath as a rail. This technique was successful in the majority of cases. The remaining patients received leads from the contralateral non-obstructed axillary vein and only one patient needed an epicardial lead. Despite that the patient cohort had a reduced left ventricular function, the complications rate was low. The Heart Rhythm Society (HRS) consensus on lead extraction in relation to lead dysfunction states that there is the option of abandoning the lead or extracting it (e.g. to reduce intravascular lead burden or regain access in the presence of venous occlusion [4]. Also, in cases of abandoned functional leads, these is an option of extracting the leads to upgrade, downgrade or replace the CIED to reduce the intravascular lead burden in order to avoid future issues. HRS states that in cases of venous obstruction or stenosis, the leads should be explanted, if the vein is going to be stented. In our study, all the patient had a degree of venous obstruction, that prevents ipsilateral lead implantation. In these patients, the obstruction is almost asymptomatic due to the development of collateral vessels. If a venous access is obtained, advancing the guidewire to bypass the obstruction/stenosis becomes the main problem. This problem can be solved by extracting a lead and after that a guidewire can be advanced through the laser sheath securing a pathway to the right atrium. The use of Lead laser extraction has shown acceptable results. In the ELECTRa study, Complete clinical and radiological success rates for transcutaneous lead extraction were 96.7% and 95.7% respectively. In-hospital procedure-related major complication rate was 1.7% including a mortality of 0.5%. and an all cause in-hospital mortality of 0.9% [9]. About 19.3% of these patients underwent laser lead extraction but the majority was performed without laser sheath which makes mechanical extraction a dominant approach. However, using the mechanical approaches is not integrated in our technique and we still have to expand our experience in this direction as it could also facilitate lead update/revision without the need for laser sheaths. On the other hand, Wazni et al has shown that laser lead extraction is highly successful with a low procedural complication rate and can be used for a wide range of indications [7]. Pokorny et al studied the difference between lead abandonment and lead extraction. They found a slight statistically non-significant difference between both strategies [10]. Bracke et al has first described the use of laser sheath to overcome venous obstruction in three patients in whom the laser sheath was advanced beyond the obstruction and the lead left in situ [11]. Gula et al also used the technique in 18 patients with no complications with a success in all cases [12]. Our study was performed in a larger patient´s cohort with expanded assessment. Two experienced operators performed the procedures in one single center. Our patients 'cohort tended to have a depressed ejection fraction and a long duration of lead implantation of 99.2 months. Most of the extracted leads were malfunctional and only 2 functional leads were extracted to be upgraded. The procedure was successful in 92.4% of the cases and intraoperative complications were limited to one patient as a superior vena cava tear which was successfully recognized and repaired. As an alternative to this procedure is the implantation of the new lead from the contralateral side or the implantation of epicardial leads. Each of these strategies carries its own drawbacks and also adding new leads is not without risk. REPLACE registry showed a 15.3% major complication rate and a 1.1% 6-month mortality rate in patients undergoing generator change with a planned lead addition or revision [13]. Increasing the leads traversing the superior vena cava can also increase the risk of superior vena cava syndrome and an increased risk of infections and erosions [14,15]. Moreover, adding new leads from the contralateral side

may induce bilateral venous occlusion. The implantation of epicardial leads could overcome the problem of venous occlusion but requires a thoracotomy and it has a limited use in cases of ICD/CRT-D [16]. Angioplasty offers an option for difficult cases but currently there is no wide spread use of this technique and outcomes data are lacking [17–19]. Antonelli et al reported their experience with supraclavicular approach to overcome ipsilateral chronic subclavian vein obstruction [20]. Lead extraction remains a complex procedure and in some cases, a very complex approach with the involvement of many specialities and multiple trials may be needed [21]. The approach is feasible and safe but there is still issues with skin irritation and lead fracture.

## Conclusion

In a single-center study on LLE in the presence of supra-cardiac occlusion of the central veins for CIED lead upgrade and revision we could demonstrate a low procedural complication rate with no procedural deaths. Most of the leads could be completely extracted to revise or upgrade the system. Our study showed a low complication rate, with acceptable mortality rates.

## Supporting information

**S1 Data.**
(XLSX)

## Author Contributions

**Conceptualization:** Sameer Al-Maisary.

**Data curation:** Sameer Al-Maisary, Gabriele Romano.

**Formal analysis:** Sameer Al-Maisary.

**Funding acquisition:** Sameer Al-Maisary.

**Investigation:** Sameer Al-Maisary.

**Methodology:** Sameer Al-Maisary.

**Project administration:** Jamila Kremer.

**Resources:** Raffaele De Simone.

**Software:** Sameer Al-Maisary.

**Supervision:** Matthias Karck, Raffaele De Simone.

**Writing – original draft:** Sameer Al-Maisary.

**Writing – review & editing:** Sameer Al-Maisary, Jamila Kremer.

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
