## [Decision Letter · Decision Letter 0]

14 Apr 2021

PONE-D-21-08523

The use of Laser lead extraction sheath in the presence of supra-cardiac occlusion of the central veins for cardiac implantable electronic device lead upgrade or revision.

PLOS ONE

Dear Dr. Sameer Al-Maisary,

Thank you for submitting your manuscript to PLOS ONE. After careful consideration, we feel that it has merit but does not fully meet PLOS ONE’s publication criteria as it currently stands. Therefore, we invite you to submit a revised version of the manuscript that addresses the points raised during the review process.

We look forward to receiving your revised manuscript.

Kind regards,

Giuseppe Coppola

Academic Editor

PLOS ONE

Journal Requirements:

4. Please upload a copy of Figure 5, to which you refer in your text on page 6. If the figure is no longer to be included as part of the submission please remove all reference to it within the text.

5. We note you have included tables to which you do not refer in the text of your manuscript. Please ensure that you refer to Tables 2 and 3 in your text; if accepted, production will need this reference to link the reader to each Table.

Reviewers' comments:

Reviewer's Responses to Questions

**Comments to the Author**

1. Is the manuscript technically sound, and do the data support the conclusions?

Reviewer #1: Yes

Reviewer #2: Yes

Reviewer #3: Yes

2. Has the statistical analysis been performed appropriately and rigorously? 

Reviewer #1: Yes

Reviewer #2: Yes

Reviewer #3: Yes

3. Have the authors made all data underlying the findings in their manuscript fully available?

Reviewer #1: Yes

Reviewer #2: Yes

Reviewer #3: Yes

4. Is the manuscript presented in an intelligible fashion and written in standard English?

Reviewer #1: Yes

Reviewer #2: Yes

Reviewer #3: Yes

5. Review Comments to the Author

Reviewer #1: The implantation of cardiac implantable electronic devices (CIED) has very increased in the last decades, reducing mortality and improvement the quality of life of patients with cardiac rhythm disorders. In this scenario the use of laser-assisted lead extraction (LLE) is a is a very useful technique, particularly in the presence of bilateral subclavian, innominate or superior vena cava obstruction, that is a major limitation to device revision and/or upgrade.

I would like congratulate all the authors contributing to this good article and research work. The article is presented in an intelligible fashion.

In conclusion, given the overall work, I recommend minor revision concerning a implementation of bibliography, reporting for example the group of Prof Curnis A, who direct a center of great experience in this field (see the paper "Curnis A, Bontempi L, et al. Active-fixation coronary sinus pacing lead extraction: a hybrid approach. Int J Cardiol. 2012 May 3;156(3):e51-2. doi: 10.1016/j.ijcard.2011.08.016").

Reviewer #2: NO COMMENT TO ADD. IT'S A GOOD REVIEW OF PATEINTS FOR A CENTER. I CAN ADD THAT IN SAME CASESE IS USEFUL TO HAVE A WIRE FROM FEMORAL VEIN TO RIGHT ATRIUM, IN CASE TOU NEED IN EMERGENCY TO INSERT A BALOON

Reviewer #3: I have read with interest the paper entitled “The use of Laser lead extraction sheath in the presence of supra-cardiac occlusion of the central veins for cardiac implantable electronic device lead upgrade or revision.”

The paper is a retrospective analysis of patients who underwent transvenous lead extraction (TLE) requiring Laser. The Authors conclude that this approach has good results in terms of safety and efficacy.

Some specific comments are:

-The section “conclusion” in the abstract is a repetition of the results and not a real conclusion. Please add that data in the section results and rearrange the section conclusion with correct form.

-It is not well described the method used for demonstration of vein occlusion (venography? CT scan, other?). Please better explain this in the method section and in the results section (how many patients performed a unilateral or bilateral venography? A CT scan?).

- It could be of interest to know how many leads had an active or a passive fixation system.

- It could be of interest the median duration of the follow up

- line 229 (discussion): you say that Lead laser extraction has shown good results. In proof of this, you report the results of the ELECTRA. It is misunderstanding, because you do not explain that only a minority of patients of the ELECTRA underwent Laser extraction (19.3%). Additionally, a sub analysis of the ELECTRA revealed that the use of mechanical dilatation was a predictor of lower incidence of major complications, compared with powered sheaths. I think that you have to explain better data from ELECTRA and extend the discussion also with an analysis of the other non-laser approach.

-In conclusion, the topic is of interest, and the paper could be suitable for the publication with the correction of these major revisions.

6. PLOS authors have the option to publish the peer review history of their article (what does this mean?). If published, this will include your full peer review and any attached files.

Reviewer #1: No

Reviewer #2: No

Reviewer #3: No

---

## [Author Response · Author response to Decision Letter 0]

2 May 2021

Rebuttal Plos one Journal

Dear Sir or Madam,

We are pleased to resubmit a major revision of the enclosed manuscript entitled: “The use of Laser lead extraction sheath in the presence of supra-cardiac occlusion of the central veins for cardiac implantable electronic device lead upgrade or revision.

” for publication in the Journal of Cardiothoracic Surgery.

We would like to express our appreciation for giving us the opportunity and the motivation to improve this paper with very valuable suggestions and elaborate comments. 

The manuscript underwent substantial revision. Details of each change to the original manuscript are reported below along with each issue raised by the Reviewers. In the manuscript, modified text passages are marked in red.

We thank you for helping us to substantially improve this paper.

Sameer Al-Maisary and Co-Authors

 

Comments from Reviewer #1:

The implantation of cardiac implantable electronic devices (CIED) has very increased in the last decades, reducing mortality and improvement the quality of life of patients with cardiac rhythm disorders. In this scenario the use of laser-assisted lead extraction (LLE) is a is a very useful technique, particularly in the presence of bilateral subclavian, innominate or superior vena cava obstruction, that is a major limitation to device revision and/or upgrade.

I would like congratulate all the authors contributing to this good article and research work. The article is presented in an intelligible fashion.

In conclusion, given the overall work, I recommend minor revision concerning a implementation of bibliography, reporting for example the group of Prof Curnis A, who direct a center of great experience in this field (see the paper "Curnis A, Bontempi L, et al. Active-fixation coronary sinus pacing lead extraction: a hybrid approach. Int J Cardiol. 2012 May 3;156(3):e51-2. doi: 10.1016/j.ijcard.2011.08.016").

Response:

We thankfully acknowledge the compliments of Reviewer 1.

We read the work of Curnis et al and found it interesting. We added it to our conclusion section to mark the complexity of some Lead extractions. We thank you again for suggesting this reference to enrich our paper.

 

Comments from Reviewer #2: 

NO COMMENT TO ADD. IT'S A GOOD REVIEW OF PATEINTS FOR A CENTER. I CAN ADD THAT IN SAME CASESE IS USEFUL TO HAVE A WIRE FROM FEMORAL VEIN TO RIGHT ATRIUM, IN CASE TOU NEED IN EMERGENCY TO INSERT A BALOON

Response:

Thank you for your comments. You have raised an important point here. We agree that inserting a wire from the femoral vein is helpful but because of the cost of the balloon we are not using it. We try to get enough financial support to be able to use it but for the time being we cannot afford it. 

 

Comments from Reviewer #3:

 I have read with interest the paper entitled “The use of Laser lead extraction sheath in the presence of supra-cardiac occlusion of the central veins for cardiac implantable electronic device lead upgrade or revision.”

The paper is a retrospective analysis of patients who underwent transvenous lead extraction (TLE) requiring Laser. The Authors conclude that this approach has good results in terms of safety and efficacy.

Some specific comments are:

-The section “conclusion” in the abstract is a repetition of the results and not a real conclusion. Please add that data in the section results and rearrange the section conclusion with correct form.

Answer:

We thank the reviewer for pointing this out. We have revised the conclusion completely. 

-It is not well described the method used for demonstration of vein occlusion (venography? CT scan, other?). Please better explain this in the method section and in the results section (how many patients performed a unilateral or bilateral venography? A CT scan?).

Answer:

This observation is correct. We have changed the sentences and made it more clear that Venography was used to make the diagnosis of venous obstruction in 23 patients. We made it also more clear in the methods section, that Venography was used for the diagnosis. We also added the word bilateral to venography as all patient with obstruction of an access vein undergo venography of the contralateral side.

- It could be of interest to know how many leads had an active or a passive fixation system.

Answer:

We thank the reviewer for pointing this out. We are sorry to tell you, that we don’t have recording of the LV-Lead type (active or passive)

- It could be of interest the median duration of the follow up

Answer:

We appreciate the reviewer’s insightful suggestion and agree that it would be useful to demonstrate that. However, trying to get these data again well need a lot of time and mostly we want be able to get the whole data. Because of that we concentrated on the first 30 days.

- line 229 (discussion): you say that Lead laser extraction has shown good results. In proof of this, you report the results of the ELECTRA. It is misunderstanding, because you do not explain that only a minority of patients of the ELECTRA underwent Laser extraction (19.3%). Additionally, a sub analysis of the ELECTRA revealed that the use of mechanical dilatation was a predictor of lower incidence of major complications, compared with powered sheaths. I think that you have to explain better data from ELECTRA and extend the discussion also with an analysis of the other non-laser approach.

Answer:

We thank the reviewer for pointing this out. We have revised our discussion and adapted your suggestions. We added that 19.3 % of the patients underwent laser lead extraction. We also added the Wazni et al paper to support the benefits of laser lead extraction to the discussion. We also discussed the role of the mechanical extraction and that we still have to improve it, to avoid the risks of laser lead extraction. 

-In conclusion, the topic is of interest, and the paper could be suitable for the publication with the correction of these major revisions.

Answer:

Thank you for your assessment. We hope that through our correction, we reach your expectations.

Once again, we thank you for the time you put in reviewing our paper and look forward to meeting your expectations. Your inputs have been precious. We look forward to hearing from you regarding our submission and to respond to any further questions and comments you may have.

The authors’

---

## [Editor Report · Decision Letter 1]

4 May 2021

The use of Laser lead extraction sheath in the presence of supra-cardiac occlusion of the central veins for cardiac implantable electronic device lead upgrade or revision.

PONE-D-21-08523R1

Dear Dr. Sameer Al-Maisary,

We’re pleased to inform you that your manuscript has been judged scientifically suitable for publication and will be formally accepted for publication once it meets all outstanding technical requirements.

Kind regards,

Giuseppe Coppola

Academic Editor

PLOS ONE

---

## [Editor Report · Acceptance letter]

6 May 2021

PONE-D-21-08523R1 

The use of Laser lead extraction sheath in the presence of supra-cardiac occlusion of the central veins for cardiac implantable electronic device lead upgrade or revision. 

Dear Dr. Al-Maisary:

I'm pleased to inform you that your manuscript has been deemed suitable for publication in PLOS ONE. Congratulations! Your manuscript is now with our production department. 

Kind regards, 

on behalf of

Dr. Giuseppe Coppola 

Academic Editor

PLOS ONE